# How do children's hospitals address health inequalities: a grey literature scoping review

Louise Brennan ,[1,2] Dora Pestotnik Stres,[1] Fiona Egboko,[1] Pallavi Patel,[1,2] Eleanor Broad,[3] Liz Brewster ,[1] Judith Lunn ,[1] Rachel Isba [1,4]

[1]Lancaster Medical School, Lancaster University, Lancaster, UK
[2]Mersey and West Lancashire Teaching Hospitals NHS Trust, Prescot, UK
[3]Dumfries and Galloway Royal Infirmary, Dumfries, UK
[4]Alder Hey Children's NHS Foundation Trust, Liverpool, UK

**Correspondence to**
Professor Rachel Isba;
rachel.isba@lancaster.ac.uk

## ABSTRACT

**Objectives** Health inequalities are systematic differences in health between people, which are avoidable and unfair. Globally, more political strategies are required to address health inequalities, which have increased since the global SARS-CoV-2/COVID-19 pandemic, with a disproportionate impact on children. This scoping review aimed to identify and collate information on how hospitals around the world that deliver care to children have addressed health inequalities.

**Design** Scoping review focused solely on grey literature.

**Eligibility criteria for selecting studies** Following Joanna Briggs Institute guidelines, a four-step approach to identifying literature was adopted.

**Data sources** Overton, OpenGrey, OpenMD, Trip Database, DuckDuckGo, Google, targeted websites and children's hospital websites were searched on March 2023 for items published since 2010.

**Data extraction and synthesis** Retrieved items were screened against clear inclusion and exclusion criteria before data were extracted by two independent reviewers using a data extraction tool. Studies were tabulated by a hospital. A meta-analysis was not conducted due to the varied nature of studies and approaches.

**Results** Our study identified 26 approaches to reduction of health inequalities, from 17 children's hospitals. Approaches were categorised based on their size and scope. Seven approaches were defined as macro, including hospital-wide inequality strategies. Ten approaches were classed as meso, including the establishment of new departments and research centres. Micro approaches (n=9) included one-off projects or interventions offered to specific groups/services. Almost half of the reported approaches did not discuss the evaluation of impact.

**Conclusions** Children's hospitals provide a suitable location to conduct public health interventions. This scoping review provides examples of approaches on three scales delivered at hospitals across high-income countries. Hospitals with the most comprehensive and extensive range of approaches employ dedicated staff within the hospital and community. This review indicates the value of recruitment of both public health-trained staff and culturally similar staff to deliver community-based interventions.

## INTRODUCTION

Health inequalities are systematic differences in health between people, which are avoidable and unfair.[1] While most

---

## STRENGTHS AND LIMITATIONS OF THIS STUDY

⇒ Unlike traditional scoping reviews of peer-reviewed literature, this review focuses solely on grey literature.
⇒ An extensive search of online grey literature databases, targeted websites and children's hospital web pages was conducted.
⇒ Grey literature may be more appropriate for areas of rapidly changing research such as new approaches to addressing health inequalities.
⇒ Items were only included if available in English or translatable through online apps.

---

commonly researched in relation to socio-economic deprivation,[2–5] the links between health inequalities and other types of social inequality such as disability,[6] race,[7] religious beliefs[8] and sexual orientation[9] are also well evidenced.

Health inequalities are evident between and within countries across the world. Two examples of this are life expectancy and under-5 mortality rates. In sub-Saharan Africa, children are 14 times more likely to die before the age of 5 than in the rest of the world.[10] In India, the under-5 mortality rate was 106 per 1000 births to mothers with no education compared with 49 per 1000 births to mothers with secondary or higher education.[11] In the UK, life expectancy is also socially patterned with, for example, people residing in the most deprived areas of England living an average of 14 years less than those in the least deprived areas.[12]

Disparities in health outcomes are also evident globally. A research conducted in the USA found that rates of low birth weight were three times higher among babies born to black mothers with no college education compared with babies born to white mothers with a college education.[13] In the UK, children living in deprived areas are more likely to suffer from conditions such as tooth decay, asthma, diabetes and obesity.[14]

However, health inequalities are not just measured in life expectancy and health outcomes; patterns in accessing care are also correlated to socioeconomic status. In New Zealand, Māori young people are the least likely to be enrolled with a primary healthcare provider.[15] In the UK, children from the most deprived areas are less likely to be brought to outpatient appointments,[16] but more likely to attend the emergency department, than those living in the least deprived areas.[17 18]

While the WHO has promoted the reduction of health inequalities as a global priority since its inaugural World Health Report in 1995,[19] political strategies to address the social determinants of health are still required in both high-income, middle-income and low-income countries.[20] In the UK, the National Health Service (NHS, a government-funded, free at the point of access, universal medical and healthcare service) has a legal duty to address health inequalities,[21] and this was made a key commitment in the *NHS Long Term Plan*.[22] However, the SARS-CoV-2/COVID-19 pandemic has widened health inequalities,[23] and critics have argued that the absence of a national health inequalities strategy has led to local healthcare systems—including hospitals—struggling to develop their own approach to addressing health inequalities.[24]

Recognising the disproportionately negative impact of the pandemic on children[25 26] and the lack of central steer on health inequalities in children and young people (CYP) in the UK[27 28] and other countries,[29] this scoping review aimed to identify and collate information on how hospitals around the world that deliver care to children have addressed health inequalities. The findings of this scoping review are intended to be used to inform children's hospitals on approaches, outcomes and experiences based on previous work, to influence future interventions and policy.

## Review questions

1. What approaches do hospitals take to address health inequalities in CYP?
2. What health inequalities do the approaches focus on?
3. How is effectiveness measured and demonstrated?

## METHODS

As presented in the published protocol,[30] this scoping review focused solely on grey literature due to the appropriateness of that approach for complex, multistakeholder interventions that lack robust data and predictability.[31] Grey literature is also more appropriate for topics that are under current review and moving at a fast pace, as is the case with interventions to address health inequalities. Furthermore, as hospitals and clinicians often have less capacity to publish academic literature in peer-reviewed journals, this methodology was deemed more appropriate as it allowed for a deeper, more systematic search of the internet and hospital websites.

### Inclusion and exclusion criteria

As detailed in the protocol, items were included if initiated by a hospital, published in English, post-2010, focused on children and with a clear aim to address health inequalities. Items were excluded if they did not explicitly state reducing health inequalities in their aim/focus and if the item did not describe an approach. Whole-hospital approaches (including adults) and antenatal and prenatal services were also excluded.

### Search strategy

In January 2023, first, the online databases Overton, OpenGrey, OpenMD and Trip Database were searched using the search terms outlined in the protocol. Second, search engines DuckDuckGo (allowing searches unaffected by previous search histories) and Google (using the anonymised search function) were searched using the search terms detailed in online supplemental appendix 1. The first 10 pages of returns were screened, unless items appeared to have continuing relevancy past 10 pages. Third, targeted websites of the American Public Health Association (APHA), WHO, NHS England, the UK's Office for Health Improvement and Disparities (OHID) and the Health Management Information Consortium (HMIC) were explored using their search functions.

Fourth, as an add-on to the original search strategy, a list of worldwide children's hospitals was identified through Wikipedia (a free online, publicly editable encyclopaedia).[32] The authors searched the full list of children's hospital's websites individually, using individual website search bars and key terms. Non-English websites were translated using the Google Translate function.

Finally, snowballing was conducted at all stages of the search, where additional resources were identified through citations or weblinks.[33]

The full search strategy and terms are detailed in online supplemental appendix 1.

### Study selection and data extraction

Searches and results were managed using Microsoft Excel. All items were screened based on their title (and where necessary associated text). Items deemed relevant were recorded by saving their website link. All items retrieved in this title screening stage were then screened in full against the inclusion and exclusion criteria by two independent reviewers. Any uncertainties or disagreements were first discussed between the two reviewers and then resolved by a third reviewer (RI). Data extraction was carried out by two separate reviewers, using a data extraction form developed using Joanna Briggs Institute guidelines[34] (online supplemental appendix 2). Where more than one approach was identified from the same hospital, they were classed as separate (and classed as a unique item) if they had separate aims. In some cases, multiple records were identified from one hospital; however, they were all related to the same aim/approach (eg, items under one strategy), and in these cases, records were combined into one entry.

## Data synthesis

Following data extraction, studies were tabulated by a hospital. Key details of the approach and its associated resources, setting and other key details were also presented alongside any outcome data. Due to the varied nature of studies and approaches, no meta-analysis was conducted. Quality assessment was not undertaken due to the grey literature focus and lack of reporting consistency.

## Patients and public involvement

No patients or public were involved.

## RESULTS

### Search results

In total, 2752 results were returned from our database searches (identification stage). Of these, 2585 were deemed inappropriate for inclusion, leaving 167 records for full-text screening against the inclusion and exclusion criteria. During full-text screening, a further 18 records were identified through snowballing. Forty-five records were identified as duplicates and excluded. Items were then excluded if they did not meet the inclusion criteria (n=108). Six records were excluded after discussion with a third independent reviewer as consensus could not be met between the initial two reviewers, leading to 115 exclusions in total. Data were extracted from 26 texts (figure 1).

### General characteristics of approaches

Articles included in the results were published online between 2017 and the search date in January 2023. The approaches were from five high-income countries (the USA, Australia, Canada, the UK and New Zealand). Seventeen separate hospitals were included in the results. Boston Children's Hospital initiated five of the included approaches, and Children's Hospital of Philadelphia (CHOP) initiated three separate approaches, with The Hospital for Sick Children (SickKids) in Toronto and Sheffield Children's Hospital each initiating two; the remainder of hospitals had one record each. A summary of the included approaches is presented in online supplemental table S1. While our protocol intended to investigate all hospitals providing care to children, the majority of our results originated from stand-alone children's hospitals.

### Categorisation of approaches

Approaches were categorised into three themes based on the size and reach of the approach they were describing and are detailed below. 'Macro' approaches were organisation-wide, impacting the activity of the whole children's hospital, for example, a formally stated strategy. The approaches were classified as 'meso' if they involved setting up a new department or would change the activity of more than one department or service. 'Micro' approaches were single interventions or projects that were short term or small in scale, for example,

reducing missed outpatient appointments. Almost half of the total 26 approaches (n=12) aimed to address a broad range of inequalities for the whole paediatric population. Mesolevel or microlevel approaches were more likely to focus on specific inequalities (eg, financial hardship[35 36]) or inequalities in access or outcomes for specific groups (eg, Aboriginal children,[37] First Nations children,[38] or refugees and asylum seekers).[39]

### Macro

Seven approaches were categorised as macro. Four of these were hospital-wide strategies, specifically aimed to address the following: health equity (Australia)[37]; equity, diversity and inclusion (EDI) (Canada)[40]; health inequalities within an innovation strategy (UK)[41]; and inclusion of health inequalities in a clinical strategy (UK).[42]

Two hospitals had an 'ethos of equity'—in Boston, this was formalised through a declaration and goals.[43] At the Starship Children's Hospital in Auckland, New Zealand, this was not formalised in a strategy but is conceptualised as a commitment in all that they do.[44]

The final macro approach to health inequalities began as a micro approach but has since become hospital-wide and replicated across the UK. The *rainbow badge initiative* is for staff, who, when signing up to wear a rainbow badge, receive information and training about the challenges that people who identify as LGBT+ can face when accessing healthcare and what they can do to support them.[45]

### Meso

Ten approaches were characterised as meso. Four of the US hospitals had established a centre aiming to address equity with dedicated staff. In Boston, the *Office of Health Equity and Inclusion* aimed to coordinate EDI initiatives across the whole hospital,[46] and the *Sandra L. Fenwick Institute* focused on research into paediatric health equity.[47] The latter has provided funding for inequalities research and delivered research projects, including an investigation into missed appointments and effective interventions to recruit diverse families. CHOP also established a *Center for Health Equity*[48] and a research *Policy Lab*, which aimed to advance health equity through a research lens.[49] The Cincinnati Children's Hospital's *Center for Child Equity* aimed to build equitable care capabilities in staff,[50] and the Seattle Children's Hospital's *Center for Diversity and Health Equity* focused on patient and family education programmes.[51] These latter hospitals have no separate research centre but include research in their equity approach.

Alongside their centre, CHOP had also established the *Healthier Together* partnership, together with community groups and non-profit government agencies to invest money into the community, families and individuals. Interventions included distribution of healthy meals, home renovation and financial counselling.[52] In Toronto, Canada, the *Family Navigation Hub* was established following the development of a social needs screening

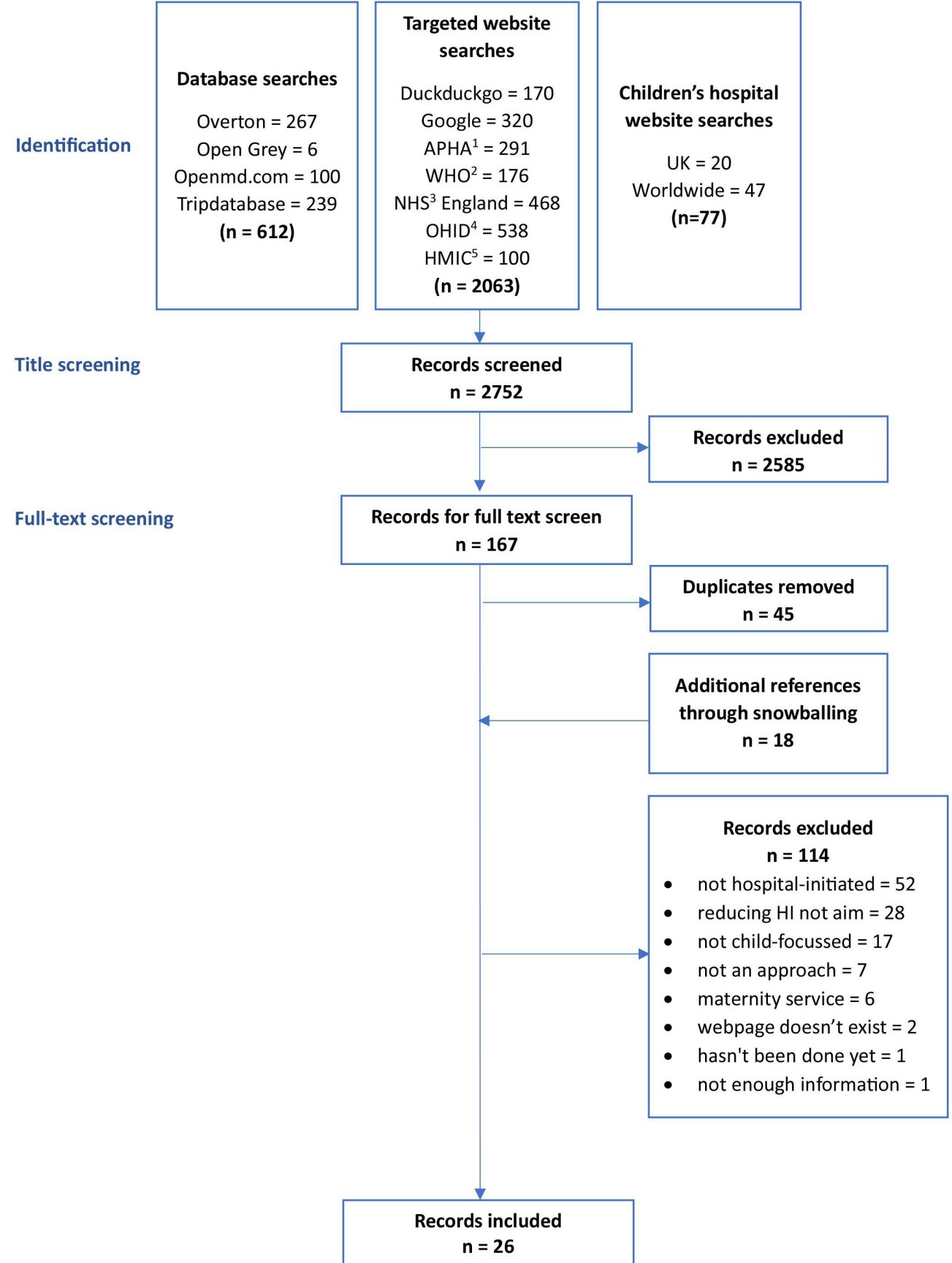

**Figure 1** Flow diagram of searches and screening. [1]APHA, American Public Health Association; [2]WHO, World Health Organization; [3]NHS, National Health Service; [4]OHID, Office for Health Improvement and Disparities; [5]HMIC, Health Management Information Consortium.

tool.[53] The hub aimed to help families to access and close the gaps around availability of housing, food, childcare, respite and community connection.[49]

Several mesolevel approaches focused on specific patient groups. An *Indigenous Liaison Service*, established in Queensland Children's Hospital, Australia, targeted support and advocacy and provided assistance to indigenous people accessing health and social services.[54] Using the nationally established *Jordan's Principle*, which ensures that First Nations children living in Canada can access services when/as they are needed, the Health Sciences Centre Children's Hospital (Winnipeg) employed a *Jordan's Principle* coordinator to assist First Nations CYP to access care at their facility.[38] Similarly, in Gateshead, England, a service had been developed to provide a single point of access for refugee and asylum-seeking children and families. This aimed to reduce barriers to healthcare by connecting families to the right health, statutory or voluntary service leading to referrals for oral health, obesity, immunisations and the tuberculosis (TB) service.[39]

## Micro

Nine approaches were categorised as 'micro', meaning they were projects or interventions used across one service/department rather than at a larger scale. Three projects originated in England. The *Families Matter* project in Sheffield aimed to help parents with financial cost of staying in hospital and provided resources to allow them to maintain basic hygiene and needs, including sanitary products, food and additional support such as signposting to additional providers of financial advice. At the time of data extraction, the hospital was piloting a free breakfast for all parents staying with their child. The feedback from staff and families was positive, with staff reporting that caregivers appeared less stressed as a result of the enhanced offers of support.[36] In Birmingham, a pilot project was running to offer free transport to families at risk of non-attendance at the hospital. Pilot data showed that 90% of the study group took up the offer of help, and 140 appointments were relocated during the 8 weeks.[35] Third, Newcastle upon Tyne Hospitals NHS Foundation Trust trialled assigning a health contact, who conducted health assessments for school-aged children living in homeless accommodation. In the 3-month pilot, 17 of the 20 children assigned a health contact had health assessments, leading to support with oral health, ophthalmology treatment, referrals for immunisations and access to the TB service.[39] Data collection and analysis were the focus of two projects that fell into the micro approach category. During the SARS-CoV-2/COVID-19 pandemic, Boston Children's Hospital analysed data on the impact of the pandemic on communities of colour and published the findings through Twitter. The posts reached more than 100 000 people, raising awareness and reaching state legislators.[55] SickKids, Canada, developed a paediatric data tool and health equity survey as part of a data collection pilot. Clerical staff collected data, and

3000 staff received training on culturally competent care. Parents and carers reported satisfaction with the data tool, and registration was identified as the optimal point in the healthcare visit to collect sociodemographic data.[56]

Community outreach projects were common in the approaches we found. In Canada, led by British Columbia Children's Hospital, the Responsive Intersectoral Child and Community Health Education and Research programme was designed to address disparities in access to primary care for children, youth and families in Vancouver's inner cities.[57] A project run by Sydney Children's Hospital aimed to improve access and equity for children with complex needs using technology and those living in rural districts. They demonstrated a 40% reduction in emergency department visits and a 42% decrease in day admissions, saving more than 50 000 km travel for families.[58] In Boston, children from low-income families of colour, with asthma, were the target population for case management and home visits by a community health worker. An 82% reduction in asthma-related hospitalisation rates was reported in the 2491 enrolled patients. Other outcomes included a 55% reduction in lost workdays for parents and a 45% reduction in missed school days.[59]

## Evaluation tools

Nearly half (n=12) of the approaches reported no evaluation/measure of success in their communication. Those hospitals reporting their outcomes used a range of different measures, with uptake/number of people involved being the most common (n=10). Several approaches used caregiver satisfaction (n=3), with qualitative findings providing powerful quotes from service users.[53] Other less common measures were the number of staff trained (n=2), accuracy of prediction tool (n=1) and money invested in the wider community by the hospital (n=1). Those hospitals that attempted to quantify their impact on health inequalities reported outcomes such as homes renovated[52] or modified,[44] change in research study uptake in diverse families,[49] reductions in hospitalisation, missed days from work and lost days at school,[59] reduction in emergency admissions and miles travelled.[58]

## Staffing/resources

Approaches varied in their approaches to staffing. Shorter-term projects used existing staff such as clerical staff for data collection,[35 56] whereas new centres/dedicated departments reported new staff structures with dedicated teams and additional infrastructure. For community outreach or targeted interventions with specific communities, positive outcomes were attributed to culturally similar staffing.[37 38 44 54] Taking the service out into the community was also reported to improve outcomes for traditionally marginalised patients, and in these cases, community health workers were often used. Approaches did not specify the cost of establishing and delivering their approach, and therefore, data could not be compared.

## DISCUSSION

In this paper, we have provided a synthesis of the approaches taken by children's hospitals in addressing health inequalities experienced by CYP. Our search of grey literature aimed to be a comprehensive way to investigate unpublished literature produced by hospitals and their partners. While our search aimed to investigate all hospitals providing care for children, the results originated solely from children's hospitals.

Our study identified 26 approaches to the reduction of health inequalities, from 16 children's hospitals, all in high-income countries. Approaches have been categorised according to their depth and breadth of reach, from macro whole-organisation approaches down to small micro pilot projects. The majority of approaches were categorised as meso and were commonly newly established departments or centres dedicated to addressing equity and research into this area.

Twelve approaches had no specific inequality as their focus, instead taking a broad approach to addressing equity. Seven hospitals addressed inequalities in access to healthcare, a further four focused on social determinants of health, and the remaining three addressed health outcomes for specific groups.

Almost half of the approaches failed to report outcomes or their methods of evaluation/assessment, and very few reported costs. Although it is notoriously difficult to demonstrate the impact of public health interventions,[60] there is still a recognised need to build an evidence base around interventions that aim to reduce health inequalities and improve population health.[61] It is, therefore, important for hospitals to better evaluate and communicate their activities and findings in the future.

Few publications relating to hospital approaches to address health inequalities in the academic literature led to the refocus of this scoping review to grey literature only. Previous research has suggested that lack of time is the most common reason for non-publication of medical and health-related studies[62]; however, this scoping review highlights the merits of publishing information on websites and other non-peer-reviewed repositories of information, while also supporting the utilisation of grey literature for reviews of hospital-based interventions.

Hospitals that did evaluate their approaches varied in their evaluation methods. At the most basic level, hospitals counted numbers involved and uptake of interventions. More complex evaluations looked at changes in missed appointments and emergency admissions by demographics such as ethnicity. Other hospitals considered missed school days and days from work for parents and carers. As this is a scoping review, an assessment of study quality was not undertaken, and findings should, therefore, be interpreted with care.

Hospitals with dedicated centres and a team of staff reported a greater number of more comprehensive outcomes. Hospitals with a dedicated health inequalities focused strategy or health inequalities within their wider strategies were also more likely to have an increased level of activity. However, strategy does not always equate to outcomes, and it is, therefore, important for hospitals to consolidate strategy with projects and appropriate staff resources. Projects involving culturally similar staff or dedicated community teams reported success in reaching historically marginalised communities.

There are several reasons for the possible exclusion of approaches from middle-income and low-income countries. First, due to time and resource constraints, our search was limited to materials available in the English language. Translate apps were used where possible as all searching was completed using the internet; nevertheless, this possibly restricted findings from some sites where translation was not accurate enough to identify alternative terms for our keywords. Second, it is highly probable that the terminology used to describe health inequalities, for example, disparities and inequities, is not universally adopted around the world, and approaches that were relevant were excluded due to the way the intervention was described. This was confirmed during screening of worldwide children's hospital websites, where several potentially relevant approaches aimed to provide culturally appropriate care rather than equity in health and were, therefore, excluded. Hence, a lack of reporting on approaches does not imply a lack of activity in this review. Third, the exclusion of hospital-wide, antenatal, prenatal and adult-focused approaches may have led to the exclusion of early years and some family-centred approaches to addressing health inequalities. Similarly, the exclusion of whole-hospital strategies that include adults may explain the results only including children's hospitals. It is possible that hospital-wide strategies existed in non-freestanding children's hospitals, which also addressed inequities in children; however, these were not eligible for inclusion in this review.

The context of healthcare systems is important to consider when interpreting these results. The majority of approaches originated from US hospitals, where healthcare is not universal and hospital activity could be driven by factors different to those where healthcare is free at the point of access. Furthermore, since the Affordable Care Act of 2010, American non-profit hospitals are mandated to undertake a community needs assessment and documentation of how they are addressing needs every 3 years.[63] Similarly, in England, there has been a national drive to address health inequalities due to their inclusion in the NHS Long Term Plan,[22] and the current 'cost of living crisis' has impacted access to care[64] and fuelled activity to address the cost of hospital appointments for patients.

## CONCLUSIONS AND RECOMMENDATIONS

Children's hospitals and hospitals that care for children provide a suitable location to conduct public health interventions. This scoping review has explored and summarised the approaches that some children's hospitals in high-income countries have taken to address

health inequalities. Several children's hospitals (mainly in the USA) have advanced equity programmes of work, but this may be driven by the regulatory environment.

There was a lack of robust evaluation of the approaches, despite public health interventions requiring appropriate evaluation to enable evidence-based practice.

The lack of published literature on hospital-led approaches to health inequalities highlights the need for hospitals to promote the work that they are doing, so areas of good practice and learning can be shared and implemented. As demonstrated here, this could be via hospital websites or through networks. The dearth of peer-reviewed articles also highlights the relevance of grey literature for hospital-based research such as this. The development of further methodological guidance on grey literature-focused scoping and systematic reviews is warranted.

The findings of this review originated from a small number of high-income countries. It is possible that relevant approaches are being used elsewhere but not being published and/or are not labelled as health inequalities. Further research into this would be warranted to complete a broader summary of approaches.

Hospitals with the most comprehensive and extensive range of approaches employ dedicated staff within the hospital and community. This review, therefore, provides evidence for the value of the recruitment of both public health-trained staff and culturally similar staff to deliver community-based interventions.

**Acknowledgements** Thank you to Louise Speakman, Lancaster University, who kindly advised on the original search strategy.

**Contributors** LB, RI, LB and JL conceptualised the work. LB, DPS and FE planned the study. DPS and LB designed the data extraction tool. LB and PP ran the database and website searches. LB, DPS, FE, PP and EB searched the children's hospital websites and were also all responsible for screening all retrieved items. LB and DPS independently extracted data. Discrepancies were discussed with RI. Themes were discussed and developed by LB, DPS, LB and JL. LB prepared the original draft. LB, DPS, FE, PP and EB designed and populated the supplementary table. RI and LB critically reviewed the work and act as joint guarantors. All authors edited and approved the final version and are accountable for the accuracy and integrity of the work.

**Funding** This work was supported by the National Paediatric Accelerator Programme, now the Children's Hospital Alliance, contracted by Sheffield Children's Hospital, grant number SCH5628.

**Competing interests** None declared.

**Patient and public involvement** Patients and/or the public were not involved in the design, or conduct, or reporting, or dissemination plans of this research.

**Patient consent for publication** Not applicable.

**Provenance and peer review** Not commissioned; externally peer reviewed.

**Data availability statement** All data relevant to the study are included in the article or uploaded as supplementary information.

**ORCID iDs**
Louise Brennan http://orcid.org/0000-0003-0377-6720
Liz Brewster http://orcid.org/0000-0003-3604-2897
Judith Lunn http://orcid.org/0000-0001-9281-2126
Rachel Isba http://orcid.org/0000-0002-2896-4309

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
