## [Reviewer comments · BMJ Open]

ARTICLE DETAILS

TITLE (PROVISIONAL)	How do children's hospitals address health inequalities: a grey literature scoping review
AUTHORS	Brennan, Louise; Stres, Dora Pestotnik; Egboko, Fiona; Patel, Pallavi; Broad, Eleanor; Brewster, Liz; Lunn, Judith; Isba, Rachel

VERSION 1 – REVIEW

REVIEWER	Osei, Lindsay Centre Hospitalier de Cayenne
REVIEW RETURNED	24-Oct-2023

GENERAL COMMENTS	Congratulations for this paper on this very important topic with limited available data. I do not have anything to say in particular.
--

REVIEWER	Javalkar, Karina Boston Children's Hospital
REVIEW RETURNED	01-Dec-2023

GENERAL COMMENTS	Thank you for the opportunity to review this article. This is an interesting article demonstrating a novel approach to literature review of hospital-wide, rapidly evolving interventions using grey literature. It is also a timely review of children's hospitals' efforts to improve health equity, which has rapidly progressed over the past few years. The paper is well-written, and the previously published protocol by the authors (Reference #30) is a good supplement to describe the methods of the grey literature review. These articles together demonstrate the utility of this type of literature review for future projects, as well as allow understanding and sharing of strategies for other children's hospitals looking to implement health equity interventions. A few minor comments are below: 1. The authors have mentioned that the review was looking to identify strategies among all hospitals that care for children; however, the results of the search only included children's hospitals. In the inclusion criteria, whole-hospital approaches were excluded. Authors may consider mentioning this in the discussion as one of the reasons why only children's hospitals were identified. There may have been whole-hospital strategies, in non freestanding children's hospitals, that also addressed inequities in children (but may not have focused specifically on children but the entire population served by the hospital) that were excluded in the review.2. In the methods, instead of "online encyclopaedia", I would recommend writing "Wikipedia" as this is more easily recognized,
--

	and then define it (as a free, publicly editable online encyclopaedia for example). 3. In the results, the title "general characteristics of studies" could be changed to "general characteristics of articles" or "general characteristics of approaches" or another similar term, since the articles reviewed were not scientific studies. The reference to the published protocol is greatly appreciated in the methods section, as this approach to literature review is novel and many will want to reference this easily through a hyperlink in the finally published manuscript. The results and discussion are clearly described and well-written. The authors discuss the implications of the findings appropriately, and describe relevant limitations including the exclusion of non-high-income countries' hospitals, English-only materials, and the influence of differential health system structures. Overall, this is an excellent article of significant interest to all who care for children, at the organization and clinician levels.
--	--

VERSION 1 – AUTHOR RESPONSE

Comment 1: The authors have mentioned that the review was looking to identify strategies among all hospitals that care for children; however, the results of the search only included children's hospitals. In the inclusion criteria, whole-hospital approaches were excluded. Authors may consider mentioning this in the discussion as one of the reasons why only children's hospitals were identified. There may have been whole-hospital strategies, in non freestanding children's hospitals, that also addressed inequities in children (but may not have focused specifically on children but the entire population served by the hospital) that were excluded in the review.

Response to comment 1: Thank you for your helpful comment. A corresponding change has been made to page 10 of the manuscript.

Comment 2: In the methods, instead of "online encyclopaedia", I would recommend writing "Wikipedia" as this is more easily recognized, and then define it (as a free, publicly editable online encyclopaedia for example).

Response to comment 2: Thank you for your helpful comment. The changes have been made as suggested.

Comment 3. In the results, the title "general characteristics of studies" could be changed to "general characteristics of articles" or "general characteristics of approaches" or another similar term, since the articles reviewed were not scientific studies.

Response to comment 3: Thank you for your helpful comment. The title has been changed to "general characteristics of approaches" as suggested.